

# STP4: spatio temporal path planning based on pedestrian trajectory prediction in dense crowds

Yuta Sato[1,2], Yoko Sasaki[1] and Hiroshi Takemura[1,2]

[1] AIRC, National Institute of Advanced Industrial Science and Technology (AIST), Tokyo, Koto-ku, Japan
[2] Department of Mechanical Engineering, Tokyo University of Science, Chiba, Noda-shi, Japan

## ABSTRACT

This article proposes a means of autonomous mobile robot navigation in dense crowds based on predicting pedestrians' future trajectories. The method includes a pedestrian trajectory prediction for a running mobile robot and spatiotemporal path planning for when the path crosses with pedestrians. The predicted trajectories are converted into a time series of cost maps, and the robot achieves smooth navigation without dodging to the right or left in crowds; the path planner does not require a long-term prediction. The results of an evaluation implementing this method in a real robot in a science museum show that the trajectory prediction works. Moreover, the proposed planning's arrival times is 26.4% faster than conventional 2D path planning's arrival time in a simulation of navigation in a crowd of 50 people.

# INTRODUCTION

Robots are required to operate in a wide range of human living spaces. Crowded environments, such as train stations and stadiums, are one of such spaces. Therefore smooth motion in dense crowds is a necessary technology for robots to be widely used in human living spaces.

Many autonomous mobile robots plan paths and motion to avoid collisions only using currently observed information, but to realize smooth motion, the path planner needs to consider the future trajectories of pedestrians. For example, the dynamic window approach (*Fox, Burgard & Thrun, 1997*) usually assumes that all currently observed obstacles remain stationary; consequently, the robot may try to move into areas occupied by pedestrians in the future, in which case it would have to freeze or make a sharp turn and start fresh to the goal. Another example is the deep reinforcement learning motion planner (DRL planner), which implicitly considers the trajectories of pedestrians (*Chen et al., 2019*; *Liu et al., 2021*; *Dugas et al., 2021*). These methods formulate the crowd navigation task as a sequential decision-making problem in a reinforcement learning framework. *Liu et al. (2021)* propose a decentralized structural recurrent neural network (DS-RNN) that understands spatial and temporal relationships in crowd navigation. In DS-RNN, time-series data is stored as a hidden state vector in the RNN, and the hidden state vector

Corresponding author
Yuta Sato, satou.mrmr@aist.go.jp

is an input to the controller. *Dugas et al. (2021)* proposed NapRep, an end-to-end DRL planner that outputs velocity commands using LiDAR raw data or local maps as inputs. NavRep combines auto-encoders and transformers to store the robot's local environment as a hidden state vector that is an input to the controller. Implementing a controller that can handle all situations with hidden state inputs takes enormous training time. In addition, the DRL planner only determines the next action and is not helpful for long-term (*i.e.,* large data size) motion planning. On the other hand, there is also a long-term planning approach in the field of multi-robots. Time enhanced $A^*$(TE$A^*$) (*Santos et al., 2015*) is a multi-robot path planning using future occupied states; it determines the search priorities of multi-robots and allows each robot to know the area it will occupy at each point. In crowds, pedestrians do not inform their future position to a robot, so long-term planning in crowds needs a framework that pairs a pedestrian trajectory prediction with spatiotemporal path planning.

There are several difficulties in incorporating pedestrian trajectory prediction into robot planning. Data-driven pedestrian trajectory prediction has been actively studied recently, and its accuracy has improved remarkably (*Alahi et al., 2016*; *Gupta et al., 2018*; *Shi et al., 2021*; *Mangalam et al., 2021*; *Yue, Manocha & Wang, 2022*). However, these prediction methods assume the input of bird's eye view information and not the input of robot observations. To combine prediction and planning, our method does not use a bird's eye view as the input but an observation by a mobile robot. As such, there are two problems with changing perspective: the first is that the mobile robot's motion affects the pedestrian trajectories. Second, the mobile robot cannot observe all pedestrians in the field. We experimentally verified the effect of these differences on the accuracy of the prediction.

This article proposes a spatiotemporal path planning based on a pedestrian trajectory prediction called STP4. STP4 can determine its behavior over the long term by explicitly combining planning with prediction. Our main contributions are:

(1) to verify how well a running robot can predict the pedestrian trajectory;

(2) to show how effective spatiotemporal path planning is for navigating in dense crowds.

In the evaluation section, we evaluate the arrival times of 2D-$A^*$, and STP4 and qualitatively assess the paths in an environment with 50 pedestrians reproduced by ORCA (*Van Den Berg et al., 2011*).

# CROWD NAVIGATION BASED ON PEDESTRIAN TRAJECTORY PREDICTION

Figure 1 is an overview of STP4. STP4 is a method that combines pedestrian trajectory prediction and spatiotemporal path planning. Our method generates a prediction map by making trajectory predictions based on the mobile robot's observations. The robot can plan paths that do not make sharp turns by searching the prediction map spatiotemporally.

## Pedestrian trajectory prediction by a mobile robot

Figure 2 shows the system flow of the proposed pedestrian trajectory prediction by a mobile robot. First, the robot's self-position and posture are estimated (Localization) in

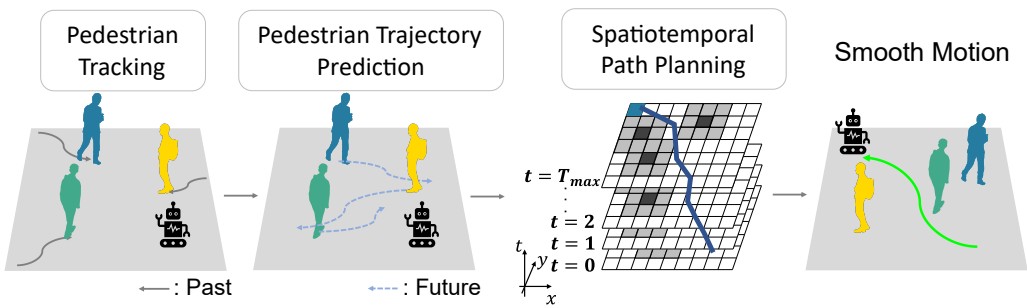

**Figure 1** Smooth motion in dense crowds considering future occupied spaces.

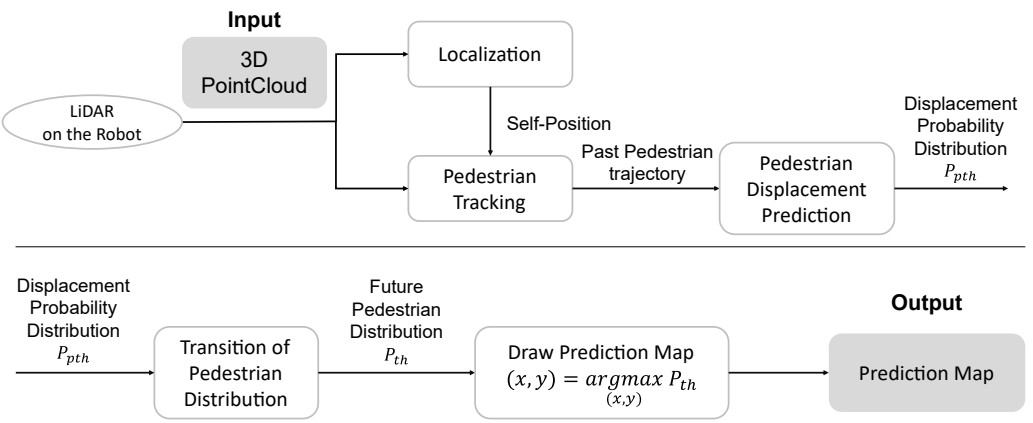

**Figure 2** The flow of pedestrian trajectory prediction by a mobile robot.

the world coordinates from 3D LiDAR input. Next, the pedestrian tracking observations in the robot's coordinate system are transformed into world coordinates (time $t$, pedestrian ID $h$, position $(x, y)$) by using self-position. Hereafter, the time $t$ will be treated as meaning the future time steps, with $t = 0$ being the current time. Next, the sparse graph convolution network for pedestrian trajectory prediction (SGCN) (*Shi et al., 2021*), a data-driven predictor, predicts the displacement of each pedestrian at each step by using their past trajectories. Finally, the transition of the pedestrian distribution is updated using the current pedestrian position and the displacement probability distribution at each step, and the future pedestrian distribution is obtained in the world coordinates system.

We use a 6-DOF Monte Carlo localization (MCL) method with 3D LiDAR and 6-axis IMU (*Hasegawa et al., 2017*). Using the displacement calculated from the IMU data as the starting point for the position and orientation estimation, the position and orientation are probabilistically obtained using a particle filter. MCL is computationally demanding because it calculates many nearest-neighbor distances between the LiDAR point cloud and the 3D map. Thus, to reduce the computational burden when the robot is on the move, the 3D map is converted in advance into a memory lookup table, in which each voxel stores the nearest-neighbor distance, thereby enabling a real-time computation.

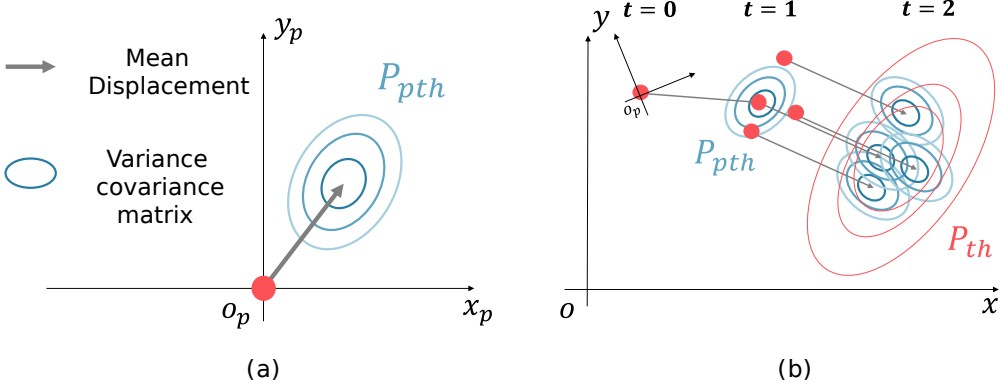

**Figure 3 Generation of future pedestrian distribution by sampling.** (A) Probability distribution of a displacement. (B) Transition of the pedestrian distribution

We use mo-tracker, a method based on multiple-hypothesis tracking (*Hatao & Kagami, 2015*) for pedestrian tracking. Mo-tracker detects registered moving object candidates using a single line of horizontal LiDAR points and calculates the trajectories of objects as a time series of positions $(x_r, y_r)$ in the robot's coordinate system. First, the support vector machine (SVM) detects and identifies the type of mobile object and its combination from the scanlines. Then, the center-of-gravity positions of the object candidates are tracked using sample-based joint probabilistic data association filters (SJPDAFs) on the time-series data. The method is robust to the approach and intersection of moving objects because it probabilistically handles multiple candidates for individual moving object detection and time series tracking for scanlines. Mo-tracker can track 20 to 30 mobile objects in real-time with a laptop computer (Intel Core i3-5010U, 16GB Memory).

There are several methods for predicting pedestrian trajectories. For example, learning algorithms directly predict a few seconds ahead without calculating the transition of the distribution. When using the prediction for collision avoidance, it is necessary to predict the trajectories not only a few seconds ahead but also at intermediate times. Therefore, the direct prediction requires loading the model into memory at each prediction time, which can be memory-consuming. For this reason, this article proposes transitioning pedestrian distribution using SGCN's output. The proposed sampling-based approach can cover the multimodal prediction for generating a time series of pedestrian distribution.

SGCN, which is known for its high performance as a data-driven method, is used to predict the displacements of pedestrians. SGCN can infer sparse graphs useful for the prediction and uses a graph convolution network to convolve pedestrian interactions. Figure 3A shows the output of the SGCN, the displacement probability distribution, and Fig. 3B shows the transition of the pedestrian distribution.

The outputs of the SGCN are the displacement probability distributions of each pedestrian at multiple steps. The probability distribution consists of the mean displacements of each pedestrian at (Fig. 3A, gray arrow) and its variance–covariance matrix (Fig. 3A, ellipse). For example, if the prediction is made 2 s ahead at 10 Hz, 20 different displacement

probability distributions are output for each pedestrian. First, the pedestrian positions are sampled according to the pedestrian distribution, and the displacement probability distribution is shifted parallel to the sampling point. The following pedestrian distribution is generated by summing the parallel shifted displacement probability distributions. Figure 3B shows an example of a future pedestrian distribution generated two steps ahead. The red dots at $t = 0$ indicate the current pedestrian position. The candidate pedestrian positions at $t = 1$ are initially sampled (red dots) according to the displacement probability distribution at $t = 1$ (blue concentric ellipses). Next, the displacement probability distribution at $t = 2$ is shifted parallel to the sampling point at $t = 1$. Then, as a superposition of the displacement probability distribution (blue ellipses), we generate the future pedestrian distribution at $t = 2$ (red concentric ellipses). Repeating this process generates a pedestrian distribution in the world coordinates system at any time. The equations below formulate the process; the future pedestrian distribution $\mathcal{P}_t$ at time $t$ in the world coordinates system is expressed as the sum of the $\mathcal{P}_{th}$s for each pedestrian:

$$\mathcal{P}_t = \sigma_1 \sum_{h \in H} \mathcal{P}_{th} \tag{1}$$

$$\mathcal{P}_{th} = \sigma_2 \sum_{s \in S} \mathcal{P}_{pth}(x - x_{hs}, y - y_{hs}) \tag{2}$$

$$(x_{hs}, y_{hs}) \sim \mathcal{P}_{(t-1)h} \tag{3}$$

where $h$ is the pedestrian ID, $H$ is the number of pedestrians at time $t$, $S$ is the number of samplings, and $\sigma_1$ and $\sigma_2$ are normalization constants. $x_{hs}$ and $y_{hs}$ denote the pedestrian position sampled at time $t - 1$.

## Spatiotemporal path planning

In this section, we focus on spatiotemporal path planning. The prediction map searched by STP4 is a grid map with a time layer. The cells of the map are defined by three types of cost $c(l)$, as shown in Eq. (4).

$$c(l) = \begin{cases} C_o \text{ if Occupied } (d \leq r_p + r_b) \\ C_c \text{ if Caution: } (r_p + r_b \leq d \leq r_p + r_b + b) \\ C_f \text{ if Free } (r_p + r_b + b \leq d) \end{cases} \tag{4}$$

Note that $C_c < C_f$ because $c(l)$ is the denominator in Eq. (7). The occupied cell is not searched for triggered by $C_o$. $d$ is the distance of the nearest pedestrian to the cell, $r_p$ is the pedestrian radius, $r_b$ is the robot radius, and $b$ is the buffer. The pedestrian position is assumed to be the $(x, y)$ at which $\mathcal{P}_{th}$ (Eq. 2) is maximized.

STP4 adds a time dimension to 2D-$A^*$. It finds the spatiotemporal path from the start node $S$ through any node $n$ to the goal node $G$ in the current map ($t = 0$) and the prediction map. Here, let $g(n)$ be the estimated minimum cost from the start node $S$ to $n$ and $h(n)$ be

the estimated minimum cost from $n$ to the goal node $G$. The estimated cost of the shortest path at $n$, $f(n)$, is calculated as follows.

$$f(n) = g(n) + h(n) \tag{5}$$

$$g(n) = \sum_{l=S+1}^{N} D(l-1, l) \tag{6}$$

$$D(l-1, l) = \begin{cases} E(l-1, l)/c(l) & \text{if } ((l-1).x \neq l.x) \wedge ((l-1).y \neq l.y) \\ 1/c(l) & \text{if } ((l-1).x = l.x) \wedge ((l-1).y = l.y) \end{cases} \tag{7}$$

$$h(n) = E(n, G)/C_f \tag{8}$$

In Eq. (6), $l-1$ corresponds to the parent node and $l$ to the child node, $D(l-1, l)$ is the cost of one step forward, and $N$ is the total number of nodes passed from the starting node $S$ to $n$. $g(n)$ equals the sum of $D(l-1, l)$s. $E(a, b)$ is the two-dimensional Euclidean distance from node $a$ to node $b$, with the unit being one cell. In Eq. (6), $((l-1).x = l.x) \wedge ((l-1).y = l.y)$ means stop. This cost is used to prevent the robot from freezing.

Figure 4 shows an overview of STP4. $T_{max}$ is the number of prediction maps, and the yellow cell is the position of the nodes at $t = (l-1).t (1 \leq l.t \leq T_{max})$. The light-green points represent the child nodes of the yellow cell. The black and gray cells are occupied and cautioned regions. The blue cells are the goal nodes. Note that in STP4, the goal node exists in all prediction maps. In other words, the goal is not a point goal as in 2D- $A^*$, but a column-like goal extending in the time direction.

STP4 constantly searches in the next step, as can be seen from the way the searching proceeds from $t = (l-1).t$ to $t = l.t$ in Fig. 4. It can also output a path with a point to stop as it searches for the same $(x, y)$ node as in the previous step. As time advances to the next step, the prediction map is updated, and the search is repeated, starting at $t = 0$. In multi-robot path planning, such as TE$A^*$ (*Santos et al., 2015*), it is always possible to prepare the prediction map to arrive at the goal because the other robots let own robot the occupied area at each future step. However, the accuracy of a prediction over a long time is less than that of a prediction over a short time, so searching for a long time would be useless. Therefore, in STP4, when the prediction maps are insufficient, 2D-$A^*$ is used with $t = T_{max}$. The evaluation section shows that STP4 has an advantage over 2D-$A^*$ even when the number of prediction maps is insufficient.

## EVALUATION OF THE PEDESTRIAN TRAJECTORY PREDICTION

We predicted real pedestrians to examine the effect of the robot's motion on the prediction accuracy.

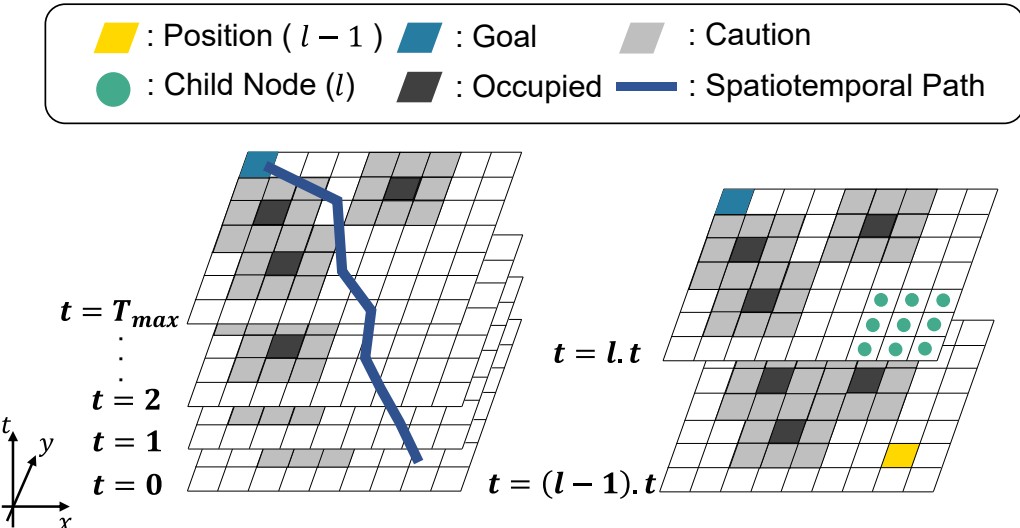

**Figure 4** Spatiotemporal path planning using the proposed prediction map.

**Table 1** Experimental settings of the pedestrian trajectory prediction.

| | |
|---|---|
| Number of pedestrians | 4 |
| Robot | WHILL Model-CR with Velodyne VLP-16 LiDAR |
| Fixed LiDAR | UTM-30LX |
| Motion capture | MotionAnalysis Eagle x 18 |
| Robot motion | Stop, Straight or Autonomous nav. |

## Experimental settings

The experiment was conducted in the laboratory with one robot and four pedestrians. The experimental settings are summarized in Table 1. Only the start and goal were given to the pedestrians. They were instructed to start moving on a cue. The robot's 3D LiDAR and a fixed LiDAR observed the pedestrian trajectory, and motion capture was used to observe the pedestrians, who wore helmets with markers, and acquire the true values for evaluation. A total of 12 trials were conducted by changing the robot's motion and the goal positions of the pedestrians.

The four starts and goals are shown in Fig. 5. In each illustration, the blue circles represent the start, and the red square represent the goal, and the symbols identify pedestrians (numbers) and robots (R). The yellow circle is the fixed LiDAR, and the blue-filled area is the motion-capture observation area (5 [m] × 5 [m]). The gray line represents the correspondence between the start and the goal. The pedestrians and the robot must pass each other.

The pedestrian trajectory data (time $t$, pedestrian ID $h$, coordinates $(x, y)$) were analyzed offline. The observation period was set to 0.1 [s], and the predictions were calculated up to 20 steps into the future (2 [s]) based on the past 20 steps. SGCN was trained on pedestrian data at the National Museum of Emerging Science and Innovation (Miraikan) (3 h 45 m,

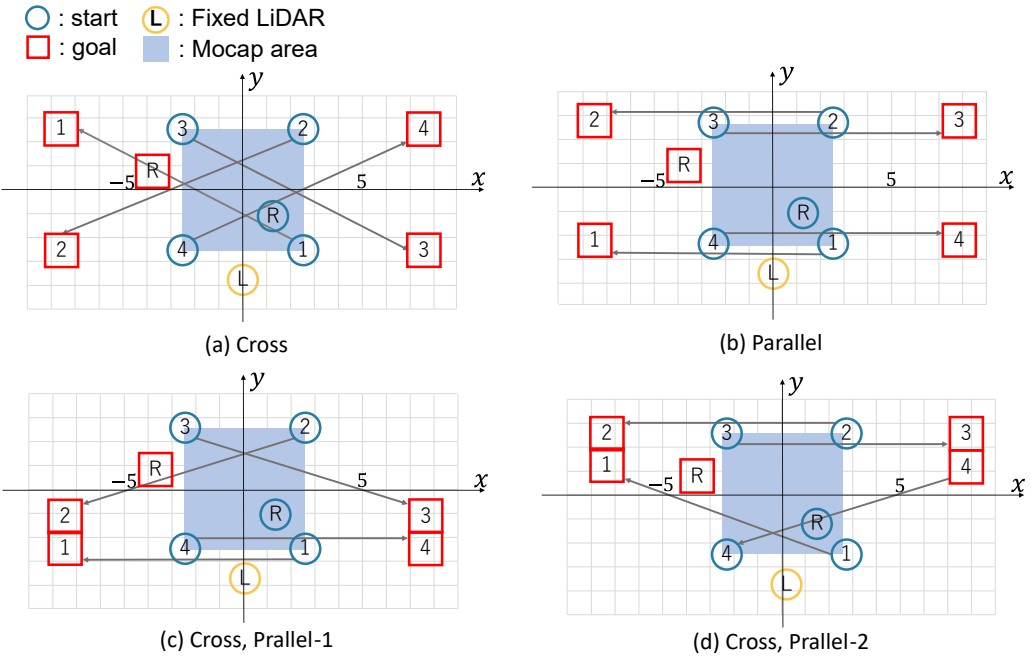

**Figure 5  Start/Goal settings for pedestrians and robots.**

21,497 trajectories). The average Euclidean error at each step (*ADE*) and the Euclidean error at the final step (*FDE*) were used as the evaluation indices (Eqs.(9), (10)). *ADE* and *FDE* are widely adopted as evaluation metrics for pedestrian trajectory prediction (*Mangalam et al., 2021*; *Yue, Manocha & Wang, 2022*). Good accuracy of *FDE* alone does not guarantee an accurate prediction at all times; it may mean that the prediction and ground truth match only at the final prediction time but not at any other time. If the accuracy of both *ADE* and *FDE* is good, it can be inferred that the similarity between the prediction and ground truth is good throughout the entire prediction time.

$$ADE = \frac{\sum_{h \in H} \sum_{t \in T_p} \sqrt{|\hat{X}_t^h - X_t^h|^2}}{H \times T_p} \tag{9}$$

$$FDE = \frac{\sum_{h \in H} \sqrt{|\hat{X_{T_p}h} - X_{T_p}h|^2}}{H} \tag{10}$$

$H$ is the number of pedestrians in the field, $T_p$ is the prediction final time, and $\hat{X}_t^h$ and $X_t^h$ denote the predicted and true coordinates of the pedestrian at time $t$ and the pedestrian ID $h$, respectively.

## Accuracy of the pedestrian trajectory prediction by the mobile robot

To verify the effect of the robot motion on the prediction, we evaluated the accuracy of the prediction for different robot motions. The tracking success rate was 93.8% for fixed

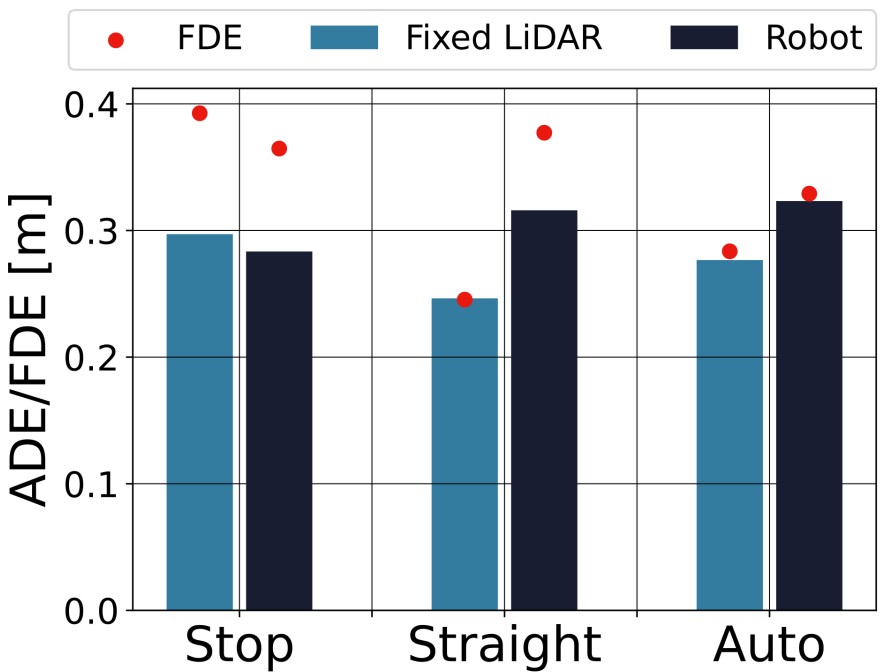

**Figure 6** **Accuracy of the pedestrian trajectory prediction for each robot motion.**

LiDAR and 60.4% for the robot. _ADE_ and _FDE_ were calculated for the successfully tracked data. The results are shown in Fig. 6. The bars indicate _ADE_, and the red dots indicate _FDE_. The light-blue bars on the left side indicate the fixed LiDAR observations, and the dark-blue bars on the right side show the robot observations.

The _ADE_s of the robot observations were not significantly different between robot motions, and their variation was comparable to the variation in _ADE_ of the fixed LiDAR. This means that the effect of the robot's motion on the prediction accuracy is small, and even if the pedestrian trajectories are affected by the robot's motion, it remains possible to predict them.

When the robot's motion was straight or auto, _ADE_ and _FDE_ were smaller for the fixed LiDAR than for the robot, whereas when the robot was stopped, there was no significant difference between the two measures. It is thought that localization errors may have affected the results when the robot was moving.

The running robot had a low success rate of tracking, but even though it could not see all of the pedestrians in the field, the effect on the prediction accuracy for successful pedestrians was small. In this experiment, the average error during 2 s was 0.3 [m], and the final error after 2 s was less than 0.4 [m], suggesting that the method can be used for collision avoidance.

## Pedestrian trajectory predictions in a science museum

We performed the proposed trajectory prediction from an autonomous mobile robot at the National Museum of Emerging Science and Innovation (Miraikan). Miraikan has

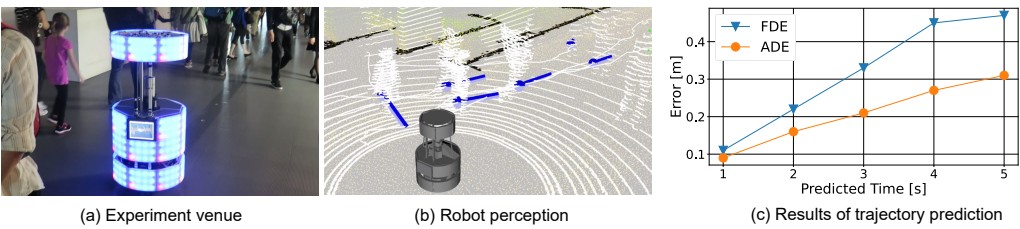

| (a) Experiment venue | (b) Robot perception | (c) Results of trajectory prediction |

**Figure 7** Future pedestrian predictions at the National Museum of Emerging Science and Innovation (Miraikan). (A) Experiment venue; (B) robot perception.

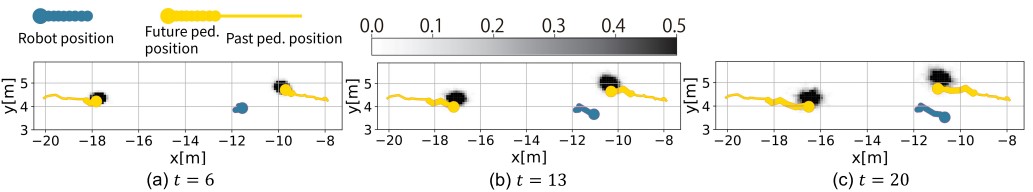

| (a) $t = 6$ | (b) $t = 13$ | (c) $t = 20$ |

**Figure 8** Examples of future pedestrian distribution during the autonomous running of the robot (Setting (C)).

approximately 1,000 to 2,000 visitors per day. The robot runs autonomously throughout the day on a floor of about 100 [m] × 30 [m]. Figure 7A shows the experiment venue, and (Fig. 7B) is a 3D view of the robot perception. The white point cloud shows the LiDAR input. Blue bars indicate the observed pedestrian position and velocity with the tracked path as a white line.

For the trajectory prediction, execution parameters are the same as the basic experiment in the lab. The trajectory prediction is evaluated with 20 min of observation, including 1,160 trajectories. Figure 7C indicates ADE and FDE up to 50 steps (5 [s]) ahead. The error at 2 [s] ahead was about half that of the basic experiment because the error is related to the robot's observed trajectory and does not include the robot's localization errors and tracking errors. FDE at 5 [s] ahead was 0.47 [m]. Given that a person's shoulder width is about 0.5 [m], this is sufficient accuracy to avoid collisions with pedestrians.

Figure 8 shows an example of the future pedestrian distribution. The observed trajectory and the future pedestrian distribution are illustrated for each of the seven steps. The black shadow represents the predicted pedestrian distribution. The solid lines are on the observed pedestrian trajectory for the past 20 steps used for input, and the thick points are the true pedestrian and robot positions. In Figs. 8B–8C the error relative to the true position increases as time advances, but it does not deviate from the direction significantly, and the distribution captures the approximate pedestrian flow. On the other hand, the proposed method could not predict pedestrians' sudden turns, such as when a person remembers that he or she forgot something.

**Table 2   Simulation settings.**

| Pedestrian | | Robot | |
|---|---|---|---|
| Number of pedestrians | 50 | Moving area range | 20 [m] × 20 [m] |
| Walking area range | 50 [m] × 50 [m] | Speed range | 0.0∼1.4 [m/s] |
| Preferred velocity | 1.0 [m/s] | Robot radius ($r_b$) | 0.1 [m] |
| Max. speed | 2.0 [m/s] | Buffer ($b$) | 0.35 [m] |
| Number of waypoints | 9 | Resolution of image | 0.05 [m] |
| Pedestrian radius ($r_p$) | 0.3 [m] | | |

# EVALUATION OF THE SPATIOTEMPORAL PATH PLANNING

Next, we evaluated the robot path planning of the proposed STP4 in dense crowds simulation. The results confirmed that STP4 has a shorter arrival time than 2D-$A^*$ and that the robot proceeds without hesitation toward the goal. The path planning code and evaluation datasets are available at https://github.com/aistairc/spatiotemporal-nav.git .

## Simulation settings

Table 2 shows the simulation settings. We set the number of pedestrians to 50 and conducted fifty trials. The gazebo simulated a crowded environment. Figure 9A shows the simulation environment. Nine waypoints (yellow circles) are set for pedestrians. The pedestrians' motions of each trial are generated as follows. The initial positions are chosen randomly from the nine waypoints and placed randomly in an 8 [m] × 8 [m] square (green square) centered on the waypoint. The distance between the pedestrians is set at least 1 [m]. The destination is randomly selected from the adjacent waypoints, and ORCA (*Van Den Berg et al., 2011*) generates their motion to the selected waypoint. When the distance to the destination is less than 3 [m], it judges the pedestrian reaching the goal and back to the goal selection process. Repeating these processes generates a crowd of people environment.

For path planning, Gazebo outputs 20 [m] × 20 [m] maps with the pedestrian trajectories (time $t$, pedestrian ID $h$, position $(x, y)$) at 20 Hz. The proposed method generates the prediction maps for each of the 20 steps (0.05[s] × 20 = 1.0[s]). In this article, $C_o = 100$, $C_c = 1$, and $C_f = 20$. Then, STP4 plans the robot's path using the prediction maps. As a comparison method, 2D-$A^*$ plans the path using the current observed map, and STP4 (True trajectory) conducts spatiotemporal planning using a perfect future map rather than the prediction maps. We evaluated the arrival time of the above three planning methods for three different Start/Goal settings. If a collision occurred, the robot stopped until the colliding obstacle was clear.

## Results

Figures 9B, 9C, and 9D show robot trajectories of the 2D-$A^*$, STP4, and STP4 (True trajectory). The robot implementing STP4 and STP4 (True trajectory) went straight ahead without hesitation toward the goal, but the one implementing 2D-$A^*$ had trouble moving. The robot trajectories of Figs. 9C and 9D are similar; the proposed (pedestrians' prediction map) is accurate enough for spatiotemporal path planning. As a quantitative evaluation of path planning, we calculate the average arrival times of the three start-goal settings for

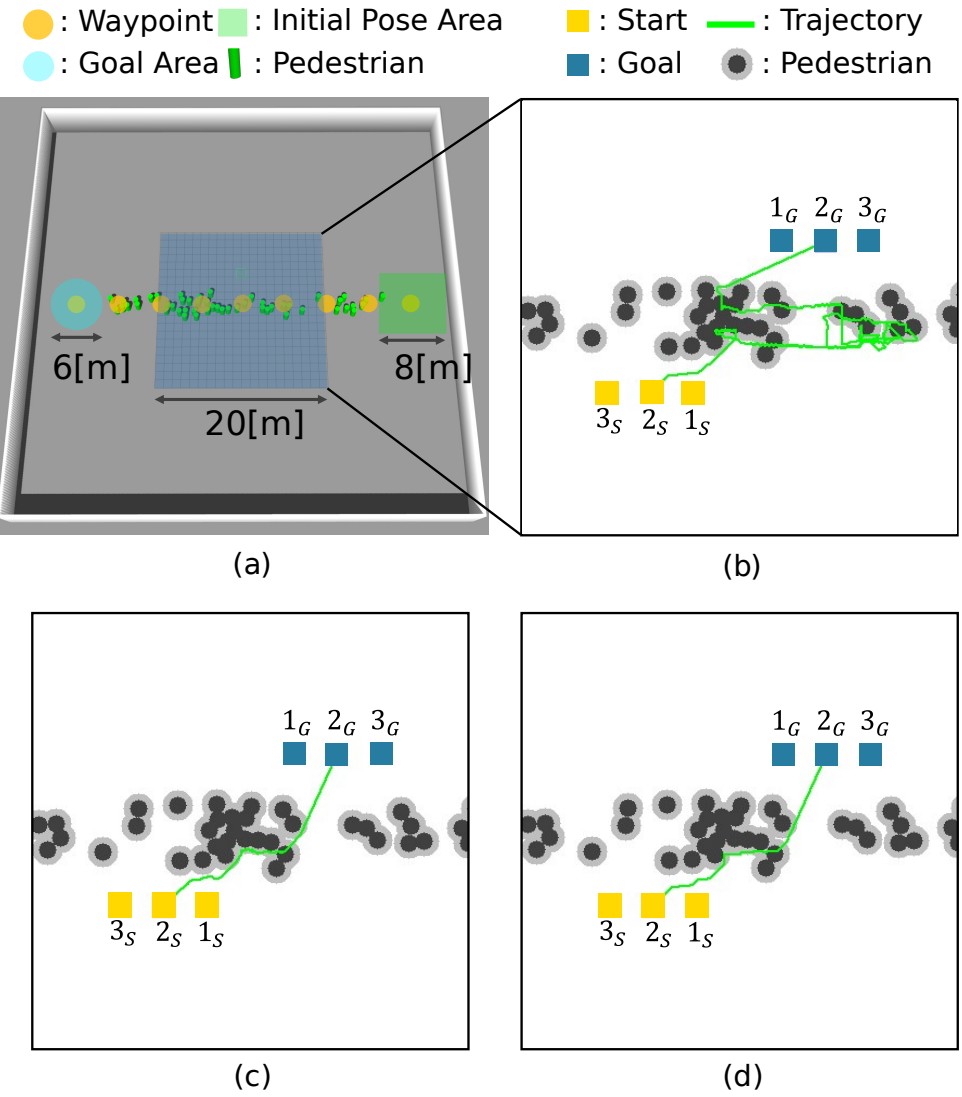

**Figure 9  Simulation environment and robot trajectories.** (A) Simulation setting. (B) 2D-*A**. (C) STP4. (D) STP4 (True trajectory).

50 scenarios. The results are shown in Fig. 10. The bars indicate the average arrival time. The blue bar represents 2D-$A^*$, the yellow bar represents STP4, and the gray bar represents STP4 (True trajectory). The error bars are standard errors.

The arrival time of STP4 (yellow bars) is 26. 4% faster than 2D-$A^*$ (blue bars) on average. The result is closer to STP4 with actual trajectory input (gray bars). The difference between the yellow and gray bars indicates the effect of prediction errors on future pedestrian distribution, and the proposed prediction of future pedestrian distribution is valid for path planning. The baseline 2D-$A^*$ only considered the shortest path at that moment, so the robot implementing it became confused whenever the situation changed, as illustrated in Fig. 9B—our spatiotemporal path planner output paths to unoccupied areas and not to

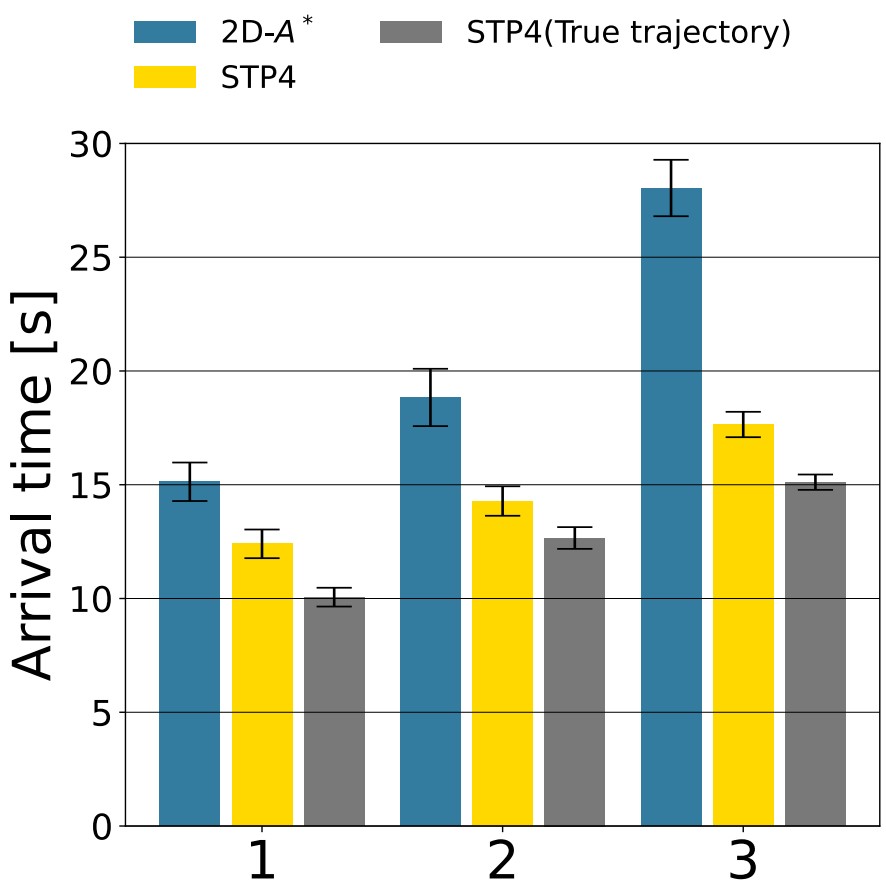

**Figure 10  The path planning evaluation of the three Start-Goal settings for 50 scenarios.**

occupied areas in the future. The robot implementing it could move without becoming confused. Looking at the arrival times for each Start/Goal combination, the percentage reduction in arrival time for 2D-$A^*$ is [1:82.0%, 2:75.8%, 3:62.9%]. The more acute the angle between the direction of human flow and the direction of robot movement, the more influential the proposed method is. The reason is that avoiding in front of or behind a person requires a more significant course change. As a result, compared to the complete map input, the arrival time of STP4 with the proposed prediction map increased by about 15%, indicating that the pedestrian prediction distribution is adequate for crowd navigation.

## CONCLUSION

The article proposed a pedestrian trajectory prediction on a running mobile robot and spatiotemporal path planning using the predicted future positions of pedestrians for autonomous navigation in crowds. Combining robot localization, pedestrian tracking, and trajectory prediction generates future pedestrian distribution. The results of an evaluation showed that the robot implementing the trajectory prediction could predict the positions of

the pedestrians within 20 steps from incomplete (2 [s]) pedestrian trajectories with similar precision as SGCN (*Shi et al., 2021*). Furthermore, the crowds' simulation of 50 people confirmed that the proposed spatiotemporal path planning using the future pedestrian map could help the robot to move smoothly, and the travel time in STP4 is 26.4% faster than the conventional method.

More work is needed to improve the prediction performance for varied pedestrians. By using behavior pattern clustering in the first stage, it is expected that the robot can deal with people approaching it, passing by, or even disturbing it.

### Funding

This work was supported by the New Energy and Industrial Technology Development Organization (NEDO). The funders had no role in study design, data collection and analysis, decision to publish, or preparation of the manuscript.

### Grant Disclosures

The following grant information was disclosed by the authors:
The New Energy and Industrial Technology Development Organization (NEDO).

### Competing Interests

The authors declare there are no competing interests.

### Author Contributions

- Yuta Sato conceived and designed the experiments, performed the experiments, analyzed the data, performed the computation work, prepared figures and/or tables, authored or reviewed drafts of the article, and approved the final draft.
- Yoko Sasaki conceived and designed the experiments, performed the experiments, prepared figures and/or tables, authored or reviewed drafts of the article, and approved the final draft.
- Hiroshi Takemura conceived and designed the experiments, authored or reviewed drafts of the article, and approved the final draft.

### Data Availability

Yoko Sasaki, National Institute of Advanced Industrial Science and Technology Artificial Intelligence Research Center, https://github.com/aistairc/spatiotemporal-nav.

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
