# Peer review of "STP4: spatio temporal path planning based on pedestrian trajectory prediction in dense crowds"

_PeerJ Computer Science, doi:10.7717/peerj-cs.1641_

## Round 0.1 · original submission · Major Revisions

The author needs to reorganize the innovation points of this paper. In the experimental part, it is necessary to clarify the experimental design, experimental results and how to prove the validity of the proposed framework. The author needs to answer each question according to the reviewer's comments and make changes in the original text.

Reviewer 1 ·

Basic reporting

This article proposes a spatiotemporal path planning method based on pedestrian trajectory prediction for a mobile robot. By utilizing mo-tracker and SGCN to predict the future positions of pedestrians, SPT4 is used for spatiotemporal path planning, enabling the mobile robot to navigate autonomously in crowds. The article is written in clear and fluent language, explaining the background and motivation well and expressing the experimental results clearly and comprehensively. Below are some suggestions to help the author further improve the paper.
1.Only one out of nine references cited in this article is from the past three years. It is recommended that the author include more recent literature to provide further evidence for the paper's novelty and to facilitate comparison with current research.
2.In the 'Introduction' section, the introduction of Time enhanced A*(TEA*) appears abrupt. It is recommended that the author provide a more detailed description of TEA* or consider revising the text to integrate this concept better.
3. In the 'Introduction' section, the author mentions, "This paper proposes a spatiotemporal path planning based on pedestrian trajectory prediction, called STP4." However, the composition of STP4 is not explicitly described in the text but is only presented in Figure 4. This may hinder readers' understanding of the paper.
4. As the innovative core point of this paper, STP4 should be highlighted throughout the text. It is suggested that the author incorporate STP4 into the title, such as "Crowd Navigation Based on Pedestrian Trajectory Prediction Using STP4" or "Spatiotemporal Path Planning with STP4".
5.The font size of some letters and numbers in Table 2, Figures 2, 4, 5, 6, and 10 are too small, and it is recommended that the author adjust the font size accordingly for improved legibility.

Experimental design

1. Please provide an explanation of why the average Euclidean error at each step (ADE) and the Euclidean error at the final step (FDE) were used as the evaluation indices in this study. Discussing the characteristics and advantages of these two evaluation metrics would be beneficial.

Validity of the findings

no comment

Reviewer 2 ·

Basic reporting

Overall, the basic background is not introduced well, where the notations are not illustrated much clear. I recommend the authors to employ certain intuitive examples to elaborate the essential notations.
What is the motivation of the proposed work? Research gaps, objectives of the proposed work should be clearly justified. The authors should consider more recent research done in the field of their study. More recent references should be added.

Experimental design

Comparsion with recent study and methods would be appreciated. Results need more explanations. Additional analysis is required at each experiment to show the its main purpose.

Validity of the findings

The author might consider justifying the performance of this study with recent study and methods.

Additional comments

I suggest that the authors add some more results. Some more theoretical Math analysis, some simulation results and some comparison of the presented scheme with other schemes. May be some figures and tables for the simulation results or the comparisons.

Reviewer 3 ·

Basic reporting

In this work, the authors proposed a framework for predicting pedestrian trajectories in dense crowds for robot navigation. Overall, the paper is well-written and organized, however, I have following comments the authors need to consider.

1. The contribution of the work is not clear. Since the proposed framework is merely a blend of existing algorithms and lacks novelty
2. The authors should critically discuss the existing methods and mention how the proposed framework fill the gaps left by existing models.

Experimental design

1. Please discuss evaluation metrics in experiment section.
2. How much similarity among predicted and ground truth trajectories is obtained by the proposed framework?
3. What are the failure cases
4. Experiment section is weak. The authors need to include more quantitative and qualitative results

Validity of the findings

1. How the authors validate the performance of proposed approach as no validatiion strategy has been adopted.

---

## Round 0.2 · accepted · Accept

The author is requested to adjust the format according to the suggestions from Reviewer 1.

Reviewer 1 ·

Basic reporting

I greatly appreciate the authors' efforts in revising the manuscript and addressing the reviewers' comments; significant improvements have been made to the paper. However, there are a few formatting suggestions: It is recommended that the bottom of Table 2 be aligned for better aesthetics. The axis labels in Figure 6 and Figure 10 appear to be overly large.

Experimental design

no comment

Validity of the findings

no comment

Reviewer 2 ·

Basic reporting

The research background and significance of this paper are clear, with good innovation.

Experimental design

The author added comparative experimental research to increase the logic of the paper, and verified the effectiveness of the proposed method.

Validity of the findings

The author added comparative experimental research, and carried out experimental research in real scenes, and obtained rich data.

Additional comments

The author has answered all the questions of the first round of review.

Reviewer 3 ·

Basic reporting

The authors have addressed all my reviews

Experimental design

The authors have addressed all my reviews

Validity of the findings

The authors have addressed all my reviews

Additional comments

The authors have addressed all my reviews